# Establishment of a Potential Serum Biomarker Panel for the Diagnosis and Prognosis of Cholangiocarcinoma Using Decision Tree Algorithms

**DOI:** 10.3390/diagnostics11040589

**Published:** 2021-03-25

**Authors:** Phongsaran Kimawaha, Apinya Jusakul, Prem Junsawang, Raynoo Thanan, Attapol Titapun, Narong Khuntikeo, Anchalee Techasen

**Affiliations:** 1Biomedical Sciences Program, Graduate School, Khon Kaen University, Khon Kaen 40002, Thailand; phongsaran.k@kkumail.com; 2Cholangiocarcinoma Research Institute, Khon Kaen University, Khon Kaen 40002, Thailand; apinjus@kku.ac.th (A.J.); raynoo@kku.ac.th (R.T.); attati@kku.ac.th (A.T.); knaron@kku.ac.th (N.K.); 3Department of Clinical Immunology and Transfusion Sciences, Faculty of Associated Medical Sciences, Khon Kaen University, Khon Kaen 40002, Thailand; 4Department of Statistics, Faculty of Science, Khon Kaen University, Khon Kaen 40002, Thailand; prem@kku.ac.th; 5Visual Intelligence Laboratory, Department of Statistics, Faculty of Science, Khon Kaen University, Khon Kaen 40002, Thailand; 6Departments of Biochemistry, Faculty of Medicine, Khon Kaen University, Khon Kaen 40002, Thailand; 7Departments of Surgery, Faculty of Medicine, Khon Kaen University, Khon Kaen 40002, Thailand; 8Department of Clinical Microbiology, Faculty of Associated Medical Sciences, Khon Kaen University, Khon Kaen 40002, Thailand

**Keywords:** cholangiocarcinoma, biomarker panel, decision tree algorithm, diagnosis

## Abstract

Potential biomarkers which include S100 calcium binding protein A9 (S100A9), mucin 5AC (MUC5AC), transforming growth factor β1 (TGF-β1), and angiopoietin-2 have previously been shown to be effective for cholangiocarcinoma (CCA) diagnosis. This study attempted to measure the sera levels of these biomarkers compared with carbohydrate antigen 19-9 (CA19-9). A total of 40 serum cases of CCA, gastrointestinal cancers (non-CCA), and healthy subjects were examined by using an enzyme-linked immunosorbent assay. The panel of biomarkers was evaluated for their accuracy in diagnosing CCA and subsequently used as inputs to construct the decision tree (DT) model as a basis for binary classification. The findings showed that serum levels of S100A9, MUC5AC, and TGF-β1 were dramatically enhanced in CCA patients. In addition, 95% sensitivity and 90% specificity for CCA differentiation from healthy cases, and 70% sensitivity and 83% specificity for CCA versus non-CCA cases was obtained by a panel incorporating all five candidate biomarkers. In CCA patients with low CA19-9 levels, S100A9 might well be a complementary marker for improved diagnostic accuracy. The high levels of TGF-β1 and angiopoietin-2 were both associated with severe tumor stages and metastasis, indicating that they could be used as a reliable prognostic biomarkers panel for CCA patients. Furthermore, the outcome of the CCA burden from the Classification and Regression Tree (CART) algorithm using serial CA19-9 and S100A9 showed high diagnostic efficiency. In conclusion, results have shown the efficacy of CCA diagnosis and prognosis of the novel CCA-biomarkers panel examined herein, which may prove be useful in clinical settings.

## 1. Introduction

Cholangiocarcinoma (CCA) is a complex group of malignancies that have arisen in the biliary tree. It is the most common liver cancer and the major public health issue of the northeastern region in Thailand [1,2]. Infection with the liver fluke *Opisthorchis viverrini*, which causes chronic inflammation and advanced periductal fibrosis, is a significant oncogenic risk factor for CCA development in this region. Additionally, the critical challenge related to this cancer is effective diagnosis and prognosis because CCA is typically asymptomatic in early stages, most often diagnosed in late stages, and difficult to differentiate from other gastrointestinal cancers (GI) including hepatocellular carcinoma (HCC) [3,4].

Currently, the blood-based tumor biomarker, carbohydrate antigen 19-9 (CA19-9), is generally used to diagnose CCA. However, the CA19-9 marker provides unsatisfactory sensitivity and specificity values because 7% of individuals in the population are Lewis antigen-negative with no or low production of CA19-9, and this marker is also often elevated in benign and other GI tract malignancies [5,6].

The numerous studies of biomarkers in bile duct cancer have focused on individual biological protein measurements (i.e., single biomarkers). There are limited studies to date concerning multiple biomarkers (biomarkers panel) that can be used as a strategy to improve the diagnostic accuracy of CCA [7,8,9]. Tshering et al. and Wongkham et al., have shown that no specific individual biomarker provides acceptable sensitivity and specificity for CCA and suggested that a combination of biomarkers may provide more accurate diagnosis [10,11]. To date, proteomics studies have been used to analyze the pattern of all proteins in a patient’s sera of many types of cancer [12,13,14]. Recent results of Duangkumpha et al. that studied the candidate proteins in sera of CCA patients compared to normal groups by mass spectrophotometry found statistically increasing levels of S100 calcium-binding protein A9 (S100A9) that had the potential to be used as the diagnostic biomarker of CCA patients [15]. A meta-analysis revealed that the detection of mucin 5AC (MUC5AC) in sera could be used as a powerful biomarker of CCA by providing a specificity of up to 97% and sensitivity of 63% [16,17,18]. Furthermore, Kimawaha et al. significantly found that transforming growth factor β1 (TGF-β1) in sera could be a potential biomarker to predict the risk of developing CCA [19]. In addition, studies of angiopoietin-2, which provides sensitivity of 74% and specificity of 94% for CCA diagnosis, have also been reported [10,11,20]. 

Currently, a computer-based diagnostic algorithm, such as decision tree classification, to construct supervised models with integrated diagnostics from candidate biomarkers, have been used for many cancers [21,22,23]. The main task of these diagnostic models is to classify the unknown objects into pre-defined groups consisting of a hierarchical structure that directs interpretation to provide a final decision [24]. Consequently, combined biomarker studies are required to establish the potentially effective CCA biomarker panels based on biomedical and bioinformatics fields, thereby enhancing the efficiency in CCA diagnosis.

In this study, we aimed to validate the group of reported candidate-CCA biomarkers, namely S100A9, MUC5AC, TGF-β1, angiopoietin-2, and the commonly used tumor marker, CA19-9, in the sera of CCA patients compared with healthy people and other GI cancer groups. The pattern of serum biomarkers together with the clinicopathological data of CCA patients was subsequently analyzed. Furthermore, to improve the diagnostic and prognostic efficiency in the management of CCA patients, the DT classification model was applied as a hierarchical structure of multi-biomarkers to establish the CCA biomarkers panel.

## 2. Materials and Methods

### 2.1. Patients and Serum Samples

Blood samples were collected from CCA (*n* = 40), and non-CCA patients (*n* = 40) including, hepatocellular carcinoma (HCC) (*n* = 23), CA gallbladder (*n* = 7), CA pancreas (*n* = 5) and liver metastasis (*n* = 5), from Srinagarind hospital and specimens were kept in the biobank of Cholangiocarcinoma Research Institute (CARI), Khon Kaen University, Khon Kaen, Thailand. In addition, blood samples of people who had normal ultrasonography results (normal group; *n* = 40) were collected from Ban Wa sub-distinct, Khon Kaen, Thailand. All human specimens and the protocols in this study were approved by the Human Ethics Committee of Khon Kaen University, based on the ethics of human specimen experimentation of the National Research Council of Thailand (HE611196), and informed consent was obtained from each subject before surgery.

### 2.2. The Detection of Candidate Proteins in Sera by Sandwich ELISA

A sandwich ELISA was performed to determine the candidate protein levels of S100A9, MUC5AC, TGF-β1, and angiopoietin-2. Quantitation of these proteins in sera of CCA patients and non-CCA patients were compared with the normal ultrasonography group by using a Quantikine ELISA Kit (S100A9, CSB-E11834h, Cusabio, Houston, TX, USA; MUC5AC, CSB-E10109h, Cusabio, Houston, TX, USA; TGF-β1, DB100B, R&D systems, Minneapolis, MN, USA; angiopoietin-2, DANG20, R&D systems, Minneapolis, MN, USA). According to the manufacturer’s instructions, the plate that was coated with primary antibody specific to each protein, was added to assay diluent to each well. Standard, control, and diluted samples were added to each well in duplicate and they were incubated for 2–2.5 h at room temperature with gentle shaking. After washing, the biotinylated antibody specific for each candidate protein was added for 1–2 h’ incubation time. Subsequently, streptavidin-HRP solution was added to each well for 45–60 min at room temperature. The TMB substrate solution was added and was protected from light. The reaction was stopped with hydrochloric acid and the plates were read on an ELISA reader using Magellan at the optical density (OD) of 450 nm. The results were calculated by reference to the standard curve that related to the concentration in each protein.

### 2.3. The Detection of CA19-9 in Sera by Automated ELECSYS COBAS

The immunoassay for quantitative determination of CA19-9 was performed by the electrochemiluminescence immunoassay (ECLIA) from Cobas e analyzer at Srinagarind hospital, Khon Kaen University, Khon Kaen, Thailand. In brief, the sandwich principle was generated by 10 µL of sample, a biotinylated monoclonal CA19-9 specific antibody, and a monoclonal CA19-9 specific antibody labeled with a ruthenium complex which formed the sandwich complex. Subsequently, streptavidin-coated microparticles were added and the complex was bound to the solid phase via an interaction of biotin and streptavidin. The reaction mixture was aspirated into the measuring cell where the microparticles were magnetically captured onto the surface of the electrode. The unbound substances were then removed and the application of a voltage to the electrode induced the chemiluminescent emission which was measured by a photomultiplier. The CA19-9 concentration was determined via a calibration curve in the analyzer. The measuring range was 0.60–1000 U/mL.

### 2.4. Decision Tree Construction for CCA Biomarkers Panel

Decision Tree (DT) is a tree-based classifier/hypothesis and has been widely considered as a practical decision-making technique with a simple knowledge representation [25]. DT has been applied in many medical applications [22,23,25]. The optimum combination of the candidate variables is derived through an exhaustive search of all possibilities by recursively partitioning a dataset to achieve the minimum impurity measure. 

We applied the Classification and Regression Tree (CART) algorithm [26] to construct a binary classification model based on five candidate biomarkers with corresponding optimal cut-off values as obtained in the previous section. The CART algorithm is an extension of C4.5 for supporting numerical target variable. A pseudo procedure of CART algorithm [26] can be summarized as follows:Start with the root node (t = 1).Search for a split s* among the set if all possible candidate ’s’ the give the purest decrease in impurity.Split node t = 1 into two nodes (t = 2 and t = 3) using the split s*.Repeat the split search process, by following the steps 1–3, for the obtained nodes (t = 2 and t = 3) until the tree grows fully or the stopping rules are met.

Three distinctive models for the binary classifier based on the CART algorithm, namely Model I for classifying normal and CCA, Model II for classifying normal and non-CCA, and Model III for classifying non-CCA and CCA, were built. For model construction, the relevant samples for binary classification were divided into two subsets defined as training (70%) and testing (30%) datasets. The distribution of clinicopathological data used for training and testing datasets are shown in Table 1. The CART algorithm was executed by Python with a Scikit-learn library [27]. Five biomarkers were considered as input variables consisting of S100A9, MUC5AC, TGF-β1, angiopoietin-2, and CA19-9. The input variables were transformed into 0 or 1 based on their obtained cut-off values. If any value satisfied the cut-off condition, then it was set to 1. Otherwise, the values were set to 0. To achieve the best tree’s parameters, a 5-fold cross-validation method with *GridSearchCV* criterion was employed to evaluate the performance of DT on the training dataset. Five parameters were investigated including max_depth, max_feature, min_samples_leaf, min_samples_split and criterion. The list of tree’s parameters with their candidate values is given in Appendix A (view Appendix A).

### 2.5. Statistical Analysis

The statistical analyses were performed using SPSS V.23.0 statistical software (IBM Corporation, Armonk, NY, USA). Data were represented as mean ± SD. The correlation of candidate biomarkers levels and clinicopathological parameters of CCA patients were analyzed by 2 independent samples t-tests. For non-parametric statistics, Mann–Whitney tests were performed. The log-rank test was used to compare survival distributions between the low and high level in sera of each protein, and the Kaplan–Meier method was plotted for survival curves for overall survival between groups. The diagnostic performance of selected proteins was evaluated using ROC curve analysis, AUC with 95% CI, and Youden index (YI) were calculated and then the optimal cut-off OD values for proteins levels were designated to balance suitable sensitivity and specificity. Odd ratios (OR) were analyzed to predict risk score by logistic regression. Values of *p* < 0.05 were considered statistically significant.

## 3. Results

### 3.1. The Validation of Candidate Biomarkers in Sera of CCA Patients

Serum levels of S100A9, MUC5AC, TGF-β1, angiopoietin-2, and CA19-9 were examined in 40 patients diagnosed with CCA, 40 non-CCA subjects, and 40 healthy individuals. The CCA sera in this validated study were obtained from 13 females (33%) and 27 males (67%). The median age of patients was 62 years (range, 39–82 years) (Table 1 and Appendix A).

As shown in Figure 1, the average values for each marker in the CCA patients were significantly higher than the normal control group, except for angiopoietin-2. Furthermore, the serum levels of S100A9 and TGF-β1 were significantly greater in the non-CCA group compared with normal subjects. Only CA19-9 levels were considerably higher in CCA patients compared with other GI cancers patients (*p* = 0.0007). 

According to subgroups of non-CCA subjects (view Appendix A), the levels of S100A9 and TGF-β1 were significantly higher in CA gallbladder and liver metastasis when compared with the normal group. Only CA19-9 was significantly higher in CCA subjects when compared with HCC and CA gallbladder patients.

### 3.2. The Correlation between Serum Candidate Biomarkers Level with Clinicopathological Data of CCA Patients

The correlations between each biomarker level and clinicopathological data of CCA patients were evaluated. As shown in Table 2, high angiopoietin-2 and TGF-β1 levels were significantly correlated with late TNM stages (stage III-IV) (OR = 4.846, 5.333; *p* < 0.05) and only a high TGF-β1 level was associated with metastasis (OR = 3.467; *p* = 0.024) and lymph node metastasis (OR = 3.111). Alternatively, there was no significant correlation of S100A9, MUC5AC, and CA19-9 levels with any clinicopathological variables.

### 3.3. The Overall Survival Analysis of Candidate Biomarkers in Sera of CCA Patients

The overall survival (OS) analysis by the Kaplan–Meier method with a log rank test revealed that only CCA patients with a high level of angiopoietin-2 showed a trend of shorter survival times than those with a low angiopoietin-2 level (*p* = 0.083). The mean overall survival times between low and high levels of angiopoietin-2 in CCA patients were 330 and 219 days, respectively (Figure 2). Nevertheless, the combination of biomarkers panel could predict poor prognosis in CCA patients. The OS analysis indicated that CCA patients with a high level of four to five combined biomarkers were found to have significantly shorter survival times than those with other levels of markers (green and yellow lines, *p* = 0.018). The mean overall survival time in each group was 445 days for all low markers, 257 days for one high marker, 317 days for two high markers, 478 days for three high markers, 280 days for four high markers, and 49 days for five high markers (view Appendix A).

### 3.4. The Combination of Candidate Biomarkers to Establish the Biomarkers Panel for CCA Diagnosis by Logistic Regression Models

ROC analysis revealed that S100A9, MUC5AC, TGF-β1, or CA19-9 individually can effectively distinguish CCA patients from the normal group (73–78% sensitivity, 58–98% specificity, AUC = 0.639–0.888, YI = 0.3–0.7, *p* < 0.05). Moreover, S100A9 alone could distinguish non-CCA from normal individuals (73% sensitivity, 88% specificity, AUC = 0.832, YI = 0.6, *p* < 0.0001) and CA19-9 alone can distinguish CCA from non-CCA patients (70% sensitivity, 83% specificity, AUC = 0.716, YI = 0.5, *p* = 0.0009) (Table 3). However, none of the single biomarkers yielded more than 90% sensitivity and specificity.

The diagnostic performance of S100A9, MUC5AC, TGF-β1, angiopoietin-2, and CA19-9 combinations as a potential CCA biomarkers panel was evaluated, using logistic regression analyses to generate models. For normal vs. CCA discrimination (Figure 3A–E; left), each cut-off values of S100A9 (197.9 ng/mL), MUC5AC (104.6 ng/mL), TGF-β1 (33.42 ng/mL), angiopoietin-2 (2422 pg/mL), and CA19-9 (23.34 U/mL) were generated from AUC analyses. The AUC for a model combining these five biomarkers based on their cut-off was 0.975 at sensitivity 95% and specificity 90% (YI = 0.85; *p* < 0.0001) which was equivalent to the combination of S100A9, MUC5AC, angiopoietin-2, and CA19-9 (YI = 0.85; *p* < 0.0001). Furthermore, combined S100A9, TGF-β1, angiopoietin-2, and CA19-9 had the second highest AUC value namely, 0.972 at 92.5% sensitivity and 90% specificity. Interestingly, the combination of only two biomarkers (S100A9 and CA19-9) had the fourth ranking according to YI to differentiate CCA from healthy individuals (95% sensitivity, 85% specificity, AUC = 0.949, YI = 0.8, *p* < 0.0001). The diagnostic power to differentiate CCA from normal subjects and the best-performing model was a combination of at least four biomarkers, namely, S100A9, MUC5AC, angiopoietin-2, and CA19-9 with a sensitivity of 90%, specificity of 95% and an AUC of 0.975.

When the non-CCA group was compared to the normal group (Figure 3A–E; middle), the best cut-off values for each protein were S100A9 (197.9 ng/mL), MUC5AC (128 ng/mL), TGF-β1 (22.81 ng/mL), angiopoietin-2 (1312 pg/mL), and CA19-9 (23.50 U/mL). The best model for diagnosis of non-CCA from the normal group was provided by the combination of five biomarkers (S100A9, MUC5AC, TGF-β1, angiopoietin-2, and CA19-9) which generated the highest AUC value (0.911) at a sensitivity and specificity of 82.5% and 85%, respectively. The combination of S100A9, MUC5AC, angiopoietin-2, CA19-9 (AUC = 0.906, 82.5% sensitivity, and 85% specificity), S100A9, MUC5AC, and angiopoietin-2 (AUC = 0.865, 77.5% sensitivity, and 87.5% specificity) had reduced diagnostic power. Interestingly, S100A9 alone or in combination with other markers featured best in most of the models. Moreover, the addition of S100A9 with angiopoietin-2 provided greater diagnostic power (AUC = 0.86, 90% sensitivity, and 72.5% specificity) for a combined two biomarkers panel model. Consequently, the best model for non-CCA prediction was obtained with a combination of S100A9, MUC5AC, angiopoietin-2, and CA19-9 which resulted in a sensitivity of 82.5%, specificity of 85% and an AUC of 0.906.

The best cut-off values that distinguished the CCA group from the non-CCA group (Figure 3A–E; right), of each protein were S100A9 (87.11 ng/mL), MUC5AC (90.51 ng/mL), TGF-β1 (39.98 ng/mL) angiopoietin-2 (1008 pg/mL), and CA19-9 (37.39 U/mL). The best model for CCA diagnosis to differentiate CCA from other GI cancers was a combination of all five biomarkers which provided the highest sensitivity (70%), specificity (82.5%), and AUC value (0.842). Moreover, the panel of two markers, CA19-9 and MUC5AC could effectively predict CCA from non-CCA at 70% sensitivity, 82.5% specificity, and AUC 0.806.

### 3.5. The Predictive Value of Candidate Biomarkers for Diagnosis CCA

By using the cut-off derived from the ROC analysis, the predictive ability of S100A9, MUC5AC, TGF-β1, angiopoietin-2, and CA19-9 on CCA diagnosis was investigated. Logistic regression analysis was used to determine whether the candidate biomarkers in sera could act as predictors for CCA burden. The crude and adjusted OR shown in Table 4, indicate that S100A9 (24.11, 22.5, *p* < 0.0001), MUC5AC (3.81, 3.78, *p* < 0.05), angiopoietin-2 (3.77, 5.17, *p* < 0.05), and CA19-9 (102.82, 129.44, *p* < 0.0001) could predict CCA from normal subjects in accordance with OR values. Interestingly, OR values revealed that levels of S100A9 and CA19-9 levels could reliably predict non-CCA from normal subjects *(p* < 0.0001, *p* < 0.05). Results of the crude and adjusted OR of MUC5AC, angiopoietin-2, and CA19-9 were 2.79 (*p* = 0.027), 3.26 (*p* = 0.016), 4.33 (*p* = 0.019), 4.78 (*p* = 0.015), 11 (*p* < 0.0001), 12.7 (*p* < 0.0001), and respectively showed that they can differentiate CCA from non-CCA patients.

### 3.6. The Diagnostic Accuracy of Candidate Biomarkers in CCA Patients with Low CA19-9 Levels

One diagnostic strategy to improve the accuracy of CA19-9 for CCA diagnosis may be to combine it with other relevant biomarkers. Among 40 patients who were diagnosed with CCA, 12 patients tested for CA 19-9 had levels lower than normal range (<37 U/mL). The number of patients who showed high S100A9 levels (≥ 197.9 ng/mL) among CCA patients with CA 19-9 below 37 U/mL was 10/12 cases (83%). Of 12 CCA patients with CA 19-9 low levels, 10 patients (black colored circle) were diagnosed using S100A9 (95% diagnostic yield) (Figure 4A). Secondly, TGF-β1 (≥33.4 ng/mL) provided diagnostic accuracy in 8/12 (67%) and 90% diagnostic yield of CCA patients with CA19-9-low levels (Figure 4C). When MUC5AC and angiopoietin-2 were combined with CA19-9 (Figure 4B,D), the proportions of CCA patients with CA19-9-low levels detected were in 6/12 cases (50%) for MUC5AC ≥ 104.6 ng/mL and 3/12 cases (25%) for angiopoietin-2 ≥ 2422 pg/mL. The diagnostic yields of MUC5AC and angiopoietin-2 were 85% and 78%, respectively.

### 3.7. The Analysis of Candidate Biomarkers as Potential Prognostic Biomarkers in CCA Patients

In this study, the prognostic biomarkers included all candidate biomarkers to monitor the prognosis in CCA patients. The results showed that only serum TGF-β1 levels were significantly different between non-metastatic and metastatic CCA patients (*p* = 0.011) (Figure 5A). Moreover, ROC analysis showed that TGF-β1 could be used to differentiate metastasis from non-metastasis with a cut-off of 48.8 ng/mL, which resulted in 44% for sensitivity and 91% for specificity (*p* = 0.012, AUC = 0.700, YI = 0.35) (Figure 5C, view Appendix A). 

For TNM stages, the results showed that serum TGF-β1 and angiopoietin-2 levels were significantly different between these groups (*p* < 0.05) (Figure 5B). ROC analysis showed that TGF-β1 could be used to predict late TNM stages with a cut-off of 43.6 ng/mL at 51% for sensitivity and 91% for specificity (*p* = 0.012, AUC = 0.748, YI = 0.42). Moreover, angiopoietin-2 level at cut-off 1457 pg/mL could predict severe stages of CCA in patients at 81% sensitivity and 78% specificity (*p* = 0.020, AUC = 0.758, YI = 0.59) (Figure 5D, view Appendix A). The combination of TGF-β1 and angiopoietin-2 level could improve the prognostic power to determine severe CCA stage in patients as shown in Figure 5E, view Appendix A; Appendix A (32% sensitivity and 100% specificity, *p* = 0.002, AUC = 0.842, YI = 0.002).

According to OR values, at cut-off of 48.8 ng/mL, the TGF-β1 level was a significant predictor to determine metastatic status (OR crude = 7.43, OR adjusted = 11.29; *p* = 0.017, 0.012, respectively). Furthermore, either TGF-β1, angiopoietin-2, or combined markers could serve as effective predictors for prognosis of severe cancer stage(s) in CCA patients (view Appendix A).

### 3.8. Decision Tree Construction and Their Diagnostic Performance for CCA Biomarkers Panel

Three models for binary classification problems called DT I (normal vs. CCA), DT II (normal vs. non-CCA), and DT III (non-CCA vs. CCA), were built in our study. After performing a five-fold cross-validation method with *GridSearchCV* criterion, the best tree’s parameters for each model were obtained and shown in Table 5. The DT diagrams for DT I (normal vs. CCA), DT II (normal vs. non-CCA), and DT III (non-CCA vs. CCA) were shown in Figure 6. The circles depict the biomarkers selected with their given cut-off conditions. The rectangles show the class predictive label with the percentage of correctly classified subjects in the training dataset. A comparative study was conducted with performance measures of each five single biomarkers and the DT models of three kinds of diagnosis computed from the confusion matrices based on the test dataset (Table 6). To differentiate CCA patients from the normal population, a DT model (DT I) was constructed, contacting a hierarchical structure of CA19-9 and S100A9 (Figure 6A). Three classification rules were obtained from DT I. The number of classification rules was characterized by the number of rectangles. From Figure 6A, subjects were labeled as CCA patients with 100% correct classification if CA19-9 was > 37 U/mL. Otherwise, S100A9 was applied to differentiate CCA and the normal cases. Using S100A9 < 197.9 ng/mL as a cut-off, subjects were labelled as normal cases with 96% correct classification. The diagnostic performance in the testing dataset of five single biomarkers and DT I was shown in Table 6 (normal vs. CCA). Interestingly, DT I gave the highest values (highlighted in bold typeface), in four performance measures, SN, YI, NPV, and ACC, and gave the second-highest values (identified by an underline) in PPV, compared to other single markers. 

To distinguish the non-CCA group from the normal group, the resulting DT model, DT II with a hierarchical structure of angiopoietin-2, TGF-β1, and S100A9, was shown in Figure 6B. All non-CCA cases were discriminated against by serum angiopoietin-2 values < 1312 pg/mL in the control group (correctly classified 83.33% in the training dataset). In comparison, TGF-β1, and S100A9 serial decision based on their relevant cut-off conditions could be accurately classified at an 87.5% level. In Table 6 (normal vs. non-CCA), DT II gave the highest values (bold typeface), in two performance measures, including YI, and ACC, and gave the second-highest values (identified by an underline) in SN, and NPV, compared to other single markers. 

Finally, DT III model suggested a serial decision involving TGF-β1 and CA19-9 to distinguish CCA patients from non-CCA subjects (Figure 6C). Using a serum TGF-β1 of value > 39.9 ng/mL and CA19-9 cut-off greater than 37 U/mL, CCA cases could be discriminated from the non-CCA, and correctly classified at 82%. Using TGF-β1 less than or equal to 39.9 ng/mL and angiopoietin-2 > 1008 pg/mL as a cutoff, non-CCA cases were identified and 100% correctly classified. Hence, MUC5AC could be used as a biomarker to discriminate CCA from non-CCA at a cut-off value higher than 90.5 ng/mL correctly providing a classification of 83%. Moreover, in Table 6 (non-CCA vs. CCA), DT III gave the highest values (bold typeface), in three performance measures, including YI, PPV, and ACC, and gave second-highest values noted by the underline in SN, and SP, compared to other single markers.

## 4. Discussion

We aimed to validate and evaluate already existing potential biomarkers for their applications in a diagnostic approach for CCA detection from healthy and related-GI cancers by using supervised learning algorithms. Based on previous findings, serum TGF-β1 alone could diagnose CCA patients at a cut-off of 38.54 ng/mL with adequate sensitivity and specificity. Interestingly, TGF-β1 combined with alkaline phosphatase (ALP), the routine liver biomarker, might provide a more efficient diagnosis of the disease given improved sensitivity and specificity [19]. Hence, only one biomarker test might not be appropriate to correctly diagnose the disease. Nevertheless, a comparatively limited number of studies have tested many blood-based biomarkers for CCA diagnosis. Thus, we must attempt to find the combination of biomarkers that might boost the capacity to diagnose CCA patients effectively.

Five previously verified CCA serum biomarkers, S100A9, MUC5AC, TGF-β1, angiopoietin-2, and CA19-9 have been validated in the sera from patients by antibody-based methods. It has been suggested that S100A9, MUC5AC, TGF-β1 and CA19-9 could theoretically differentiate CCA from the normal population [11,19]. However, only CA19-9 has been shown to distinguish CCA from GI cancers, especially in HCC patients as determined by its diagnostic performance [28,29].

The S100 protein family comprises a group of small acidic calcium proteins which have two major members, S100A8 and S100A9. S100A9 has emerged as an effective pro-inflammatory mediator in acute and chronic inflammation, and can play a critical role in cancer associated with inflammation [30]. Many studies have found that the serum level of S100A9 is significantly increased in many types of cancer and benign biliary diseases (BBD) [31,32,33]. These findings established that S100A9 was a promising diagnostic biomarker with 78% sensitivity, 88% specificity, and a 0.888 AUC value, which was equivalent values for the differential diagnosis of CCA and normal control [15]. When S100A9 level was combining with CA19-9 to enhance the diagnostic efficiency; the sensitivity value increased from 78% for S100A9 alone to 95% for these two markers. Impressively, S100A9 provides a diagnostic yield of 95% in CCA patients with low CA19-9 levels. These results suggest the potential diagnostic usefulness of S100A9 in combination with CA 19-9 or in cases in which the CA19-9 level is normal or low. 

MUC5AC is a high molecular weight *O*-glycosylated glycoprotein member of the membrane-bound and secreted epithelial mucin family. This is the most studied mucin with high potential as a biomarker for CCA [34]. We have shown that the serum levels of MUC5AC are greater in CCA than healthy subjects, and when two markers were combined, only MUC5AC and CA19-9 obtained the highest AUC value for differentiating CCA from GI tract cancers patients. Serum MUC5AC is a highly particular tumor-associated mucin that could be helpful in the diagnosis and development of therapeutic strategies for biliary tract cancer (BTC), as supported by a previous study [35]. In addition, the BTC tumor biopsies of most patients have demonstrated a high MUC5AC reactivity, suggesting the tumor-associated MUC5AC tumor antigen is shed into the blood where it can be detected [36]. Currently, serum extracellular vesicles (EVs) carry a lot of promising source of clinically beneficial biomarkers to increase cancer detection sensitivity and specificity. Arbelaiz et al. [37] discovered a new potential biomarker in serum EVs of CCA, primary sclerosing cholangitis (PSC), and HCC patients. According to their report, CCA-derived EV include oncogenic proteins including mucin and the S100 protein family, which have a high differential diagnostic capacity for CCA diagnosis [37]. Thailand is the endemic area of liver fluke infection, which is the major cause of CCA burden in our region. As a result, the serum EVs from liver fluke related CCA patients should be examined further in a prospective study, as they could provide a possible biomarker derived from EVs. This may have contributed to S100A9 and MUC5AC being the one reliable diagnostic marker of the CCA biomarker panel. 

Most of the mortality of CCA patients comes from poor prognosis, therefore, prognostic markers are needed to follow up the treatment outcomes after resection and to predict those who will benefit from treatment. Additionally, TGF-β1, the multifunctional polypeptides with potent effects, had the diagnostic and prognostic potential serum levels which was confirmed our previous in CCA studies [19]. Even though, in this study, sera TGF-β1 level appears to be less of a diagnostic power to differentiate CCA from the control group based on an unsatisfactory sensitivity and specificity values. However, our study revealed that serum TGF-β1 could significantly serve as the prognostic biomarker for monitoring metastasis and severe tumor stages of CCA.

Many studies have shown that elevated TGF-β1 levels are significantly associated with metastasis and poor prognosis in many cancers [38,39,40], as TGF-β1 can modulate the metastatic potential of tumor cells by regulating their ability to break down and infiltrate barriers of the basement membrane [41]. In CCA cell lines, the metastatic role of TGF-β1 was shown to effectively induce CCA cell migration by activation of the expression of Twist, N-cadherin and vimentin [42]. These results suggest that a possible prognostic biomarker for monitoring pathological conditions in patients with CCA may be TGF-β1.

In general, angiopoietin-2, an endothelial cell-specific angiogenic growth factor, has been used as an angiogenesis-related biomarker of various types of tumors, but has not been thoroughly examined for expression and function in CCA. According to our current study, angiopoietin-2 alone was not one of the best potential diagnostic biomarkers, in contrast to a previous study which revealed that the serum angiopoietin-2 level can be useful for differentiating CCA versus primary sclerosing cholangitis (PSC) with an acceptable AUC value [20]. The different controversial aspect is that we did not select the population of BBD as PSC patients. We only conducted angiopoietin-2 determination in CCA compared to those with normal and non-CCA groups, which were different from previous research examining only CCA, PSC, and bile duct stones in patients [20]. The different populations studied may provide explanations for the discrepancies between studies in the diagnostic outcomes of angiopoietin-2 in CCA diagnosis. However, a high level of angiopoietin-2 could be associated with the trend for shorter survival time and predict the severe cancer stages with an adjusted OR equivalent to 23.22. Preliminary studies indicated a potential role for angiopoietin-2 as a prognostic factor in cancers, for instance breast cancer [43], lung cancer [44], and HCC [45], not only by inducing angiogenesis but also by encouraging metastasis via the α5β1 integrin/integrin-linked kinase (ILK)/Akt, GSK-3β/Snail/E-cadherin signaling pathway [46]. Additionally, the combination of TGF-β1 and angiopoietin-2 could strongly predict the relative risk of poor prognosis in severe cancer stages in our study. The coincidence of this phenomenon could be explained because tumor angiogenesis is regulated by a network of growth factors, including members of the TGF-β family [47] and angiogenic inducers [48]. However, in-depth studies on the roles of these biomarkers in CCA genesis are required. 

The CART algorithm is based on Classification and Regression Trees by Breiman et al. [26]. A CART tree is a binary decision tree that is built by repeatedly splitting a node into two child nodes, starting with the root node holding the entire sample of learning. In this analysis, this algorithm was used as a classifier because it provides a set of rules that theoretically describe the relationship between inputs, including candidate biomarkers, and output as a diagnostic outcome; normal, non-CCA, or CCA. The Python-built DT diagram with the Scikit-learn library provides physicians with an easy and practical guideline to diagnose CCA patients without depending on any additional computers and other devices. 

In our study, because its performance reached the defined goal, DT was still preferred to the artificial neural network and CART can provide a logical rule set that is convenient for medical approach. For the training set, a five-fold cross-validation method with *GridSearchCV* criterion was employed to evaluate the performance of DT to achieve the best tree’s parameters. This provided three models consisting of various candidate biomarkers with varying accuracy, which was greater than the precision achieved by any single biomarker. Of these, for normal versus CCA, the two markers CA19-9 and S100A9 provided diagnostic power better than those of other multiple markers and better than any single marker. The diagnostic power of these two markers was further validated in the testing set and revealed the best diagnostic power to discriminate CCA from the healthy group. 

The challenging aspect of CCA diagnosis is to reliably distinguish CCA from other gastrointestinal cancers that demonstrate the disease’s similar pathophysiology. According to results from CCA versus non-CCA groups, we found that the integrated and combined analyses of novel candidate-biomarkers (TGF-β1, CA19-9, angiopoietin-2, and MUC5AC) tends to be a successful method for increasing the CCA diagnostic with adequate 82% sensitivity and 92% specificity. This is similar to Pattanapairoj and co-workers, who showed that the potential classification model consisted of CCA-CA and ALP for differentiating CCA from non-CCA [49]. Moreover, Negrini et al. studied the efficiency of machine learning models according to the plasma bile acid profiles and reported that the Naïve Bayes model demonstrated the improved diagnostic efficiency for the differentiation of patients with CCA and BBD [50]. In terms of diagnostic capacity, however, the classification output for this DT model (CCA versus non-CCA) was still not satisfactory. There is an important need for further research to explore the potential biomarker panel to differentiate CCA from other cancers, especially HCC. Recently, Jamnongkan et al. firstly identified glycoform patterns of serotransferrin in CCA serum by the glycoproteomic method. The results revealed that serotransferrin glycoform 6503, which is the highly-sialylated glycoform, could be used to differentiate CCA from HCC patients [51]. This study could provide a novel insight for the discovery of novel glyco-biomarkers for CCA diagnosis.

Our study has revealed some important features for the diagnosis of CCA. Firstly, we recruited GI-related cancers including hepatocellular carcinoma, CA gall bladder, CA pancreas, and liver metastatic patients that have the pathological conditions similar to CCA. Thus, the real diagnostic performance of the test is reflected more accurately than using cancer subjects versus normal controls alone. Secondly, our study focused on the finding that candidate biomarkers are correlated with clinicopathological data of CCA patients. Thirdly, our study was based on serum ELISA analysis that measures the concentration of biomarkers by real quantitative units. No previous studies have investigated a panel of these biomarkers by quantitative analyses. Lastly, this study provided a new DT algorithm for physicians with an easy and practical guideline as a potential workflow to diagnose CCA patients effectively. However, the study still has some limitations. In contrast with the number in the CCA group, the number of individuals in the non-CCA subgroups is small. Our findings can be further confirmed by other potential prospective trials with greater sample sizes involving patients with malignant GI diseases, and the biomarkers panel could be validated in an external bank of liquid biopsies to support the conclusion of this report.

## 5. Conclusions

The present study suggested the efficacy of utilizing combined biomarker analysis for CCA diagnosis. The DT algorithm was used to establish the CCA biomarker panel that could distinguish CCA from healthy people, with a panel consisting of CA19-9 followed by S100A9 having the highest diagnostic power. In CCA patients with low CA19-9 levels, S100A9 was especially useful and could be used as a complementary marker to provide greater diagnostic yield. For GI cancers versus CCA diagnosis, the results showed that the potent serial biomarkers model was obtained by CA19-9, followed by MUC5AC, and TGF-β1. Moreover, the set of two markers, TGF-β1 and angiopoietin-2, provided effective prognoses in CCA patients with metastasis and severe cancer stage conditions. Our results strengthen the value of a blood-based biomarkers panel for the diagnosis and prognosis of CCA and discloses the classification of a DT model that can be used as an effective tool for CCA diagnosis.

## Figures and Tables

**Figure 1 diagnostics-11-00589-f001:**
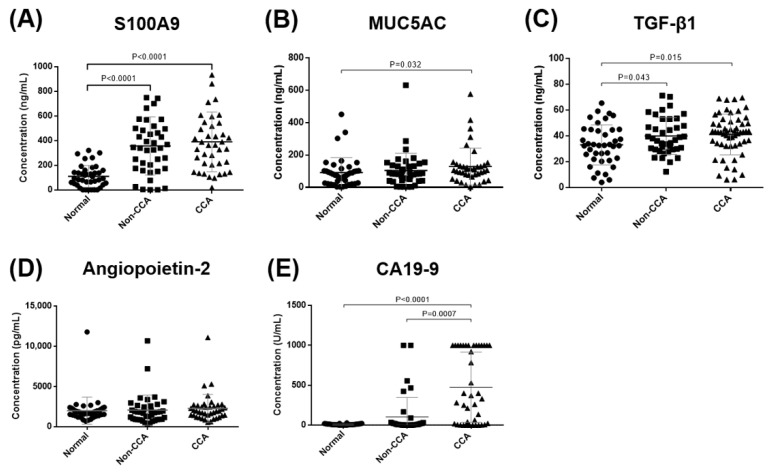
Serum levels of S100A9 (**A**), MUC5AC (**B**), TGF-β1 (**C**), angiopoietin-2 (**D**), and CA19-9 (**E**) in the normal control group, non-CCA group, and CCA patients. Scatter plots represent mean ± standard deviation (SD). The *p* value < 0.05 was considered statistically significant when compared in each group.

**Figure 2 diagnostics-11-00589-f002:**
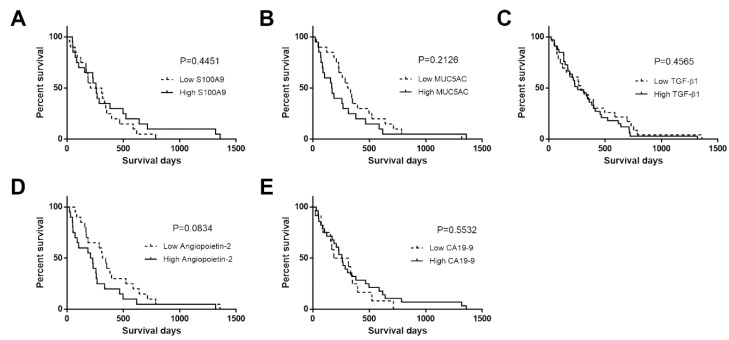
Overall survival (OS) analysis according to Kaplan–Meier method with a log rank test calculated for S100A9 (**A**), MUC5AC (**B**), TGF-β1 (**C**), angiopoietin-2 (**D**), and CA19-9 (**E**). The *p* value < 0.05 was considered statistically significant.

**Figure 3 diagnostics-11-00589-f003:**
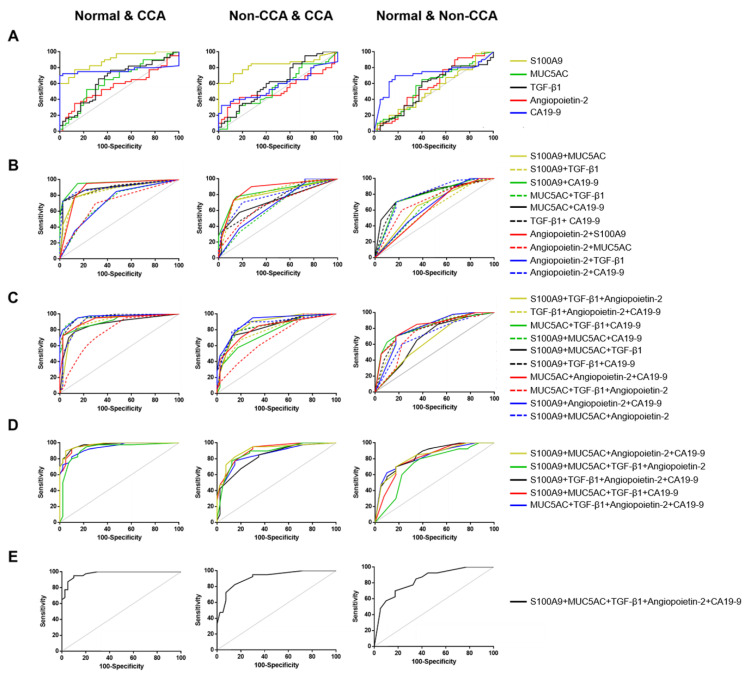
Receiver operating characteristic curve (ROC) analysis of single biomarker (**A**), two biomarkers (**B**), three biomarkers (**C**), four biomarkers (**D**), and combined five biomarkers panel (**E**) for each pairwise comparison including normal versus CCA (left), normal versus non-CCA (middle), and CCA versus non-CCA (right). Gray solid line: theoretically perfect performance of a potential biomarker as the reference line. Area under ROC curve (AUC) and statistic comparison are indicated. The *p* value < 0.05 was considered statistically significant.

**Figure 4 diagnostics-11-00589-f004:**
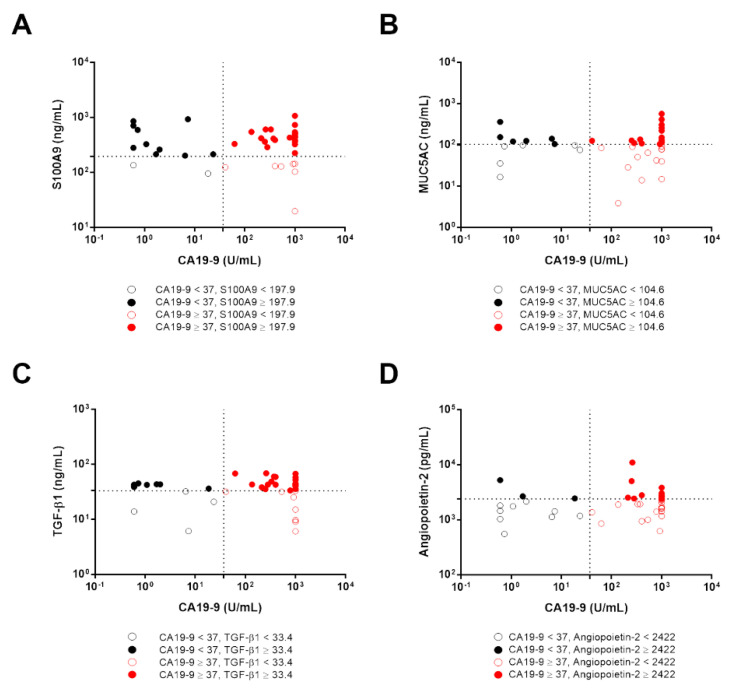
Biomarkers panel in CCA patients with CA19-9 low levels. The scatter plots indicated the distribution of CA19-9 levels and candidate biomarkers including S100A9 (**A**), MUC5AC (**B**), TGF-β1 (**C**), and angiopoietin-2 (**D**) in patients diagnosed with CCA. The optimal cut-off levels of S100A9, MUC5AC, TGF-β1, angiopoietin-2, and CA19-9 were 197.9 ng/mL, 104.6 ng/mL, 33.4 ng/mL, 2422 pg/mL, and 37 U/mL, respectively.

**Figure 5 diagnostics-11-00589-f005:**
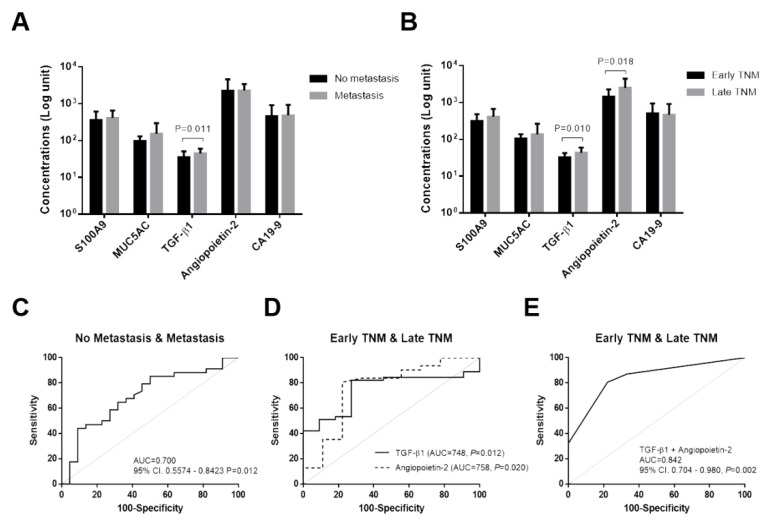
Serum levels of candidate prognostic-biomarkers in CCA patients according to metastasis (**A**) and TNM stages (**B**). Column bar graph represents mean ± standard deviation (SD). Receiver operating characteristic curve (ROC) analysis of candidate biomarkers for prognosis in CCA, TGF-β1 for prediction of metastasis (**C**), TGF-β1 and angiopoietin-2 for prediction of TNM stages (**D**), combined TGF-β1 and angiopoietin-2 for prediction of TNM stages (**E**). The *p* value < 0.05 was considered statistically significant.

**Figure 6 diagnostics-11-00589-f006:**
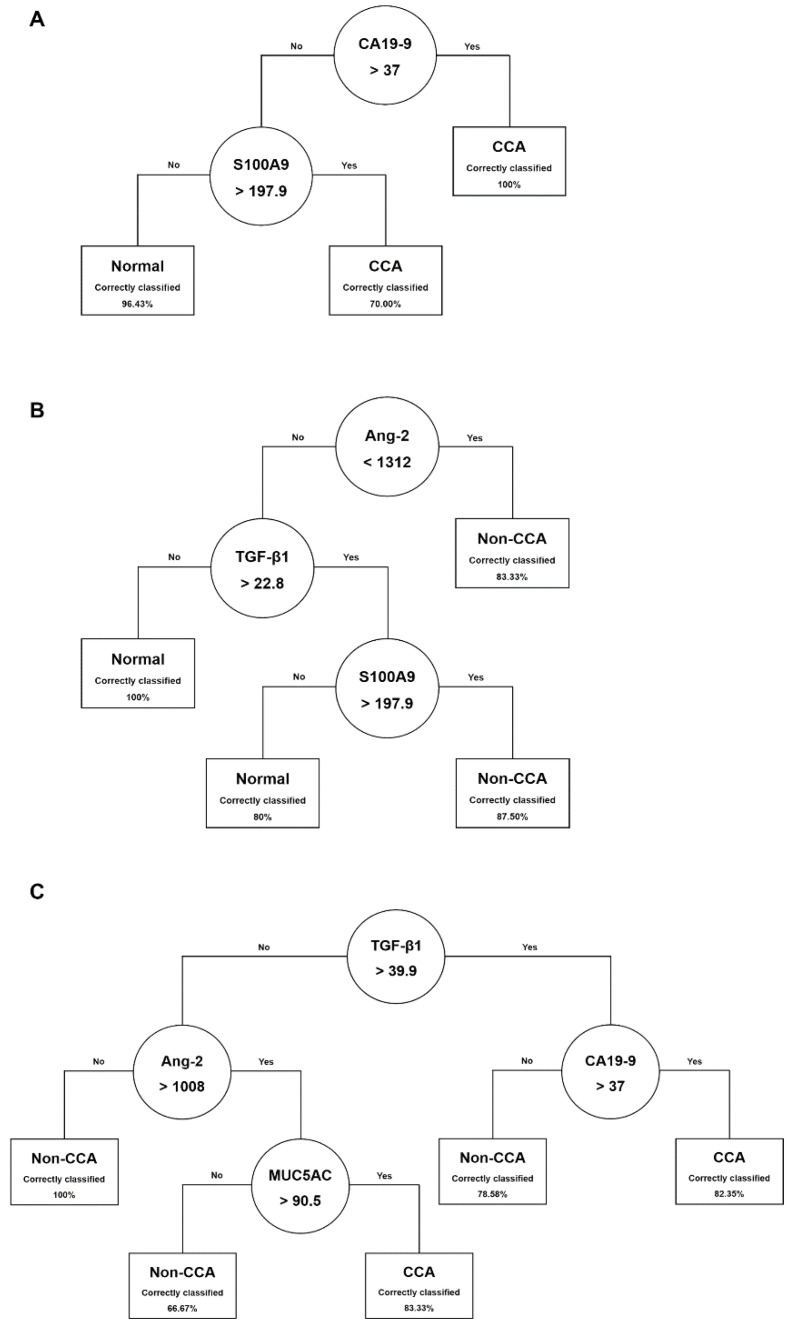
Decision tree (DT) models to classified CCA patients from healthy subjects (**A**), non-CCA patients from normal (**B**), and CCA patients versus non-CCA group (**C**) in the dataset used in this study. Establishing workflow potentially used in CCA diagnosis.

**Table 1 diagnostics-11-00589-t001:** Demographics and clinicopathological data of studied cohort.

Variables	Normal	Non-CCA				CCA
		HCC	CA Gallbladder	CA Pancreas	Liver Metastasis	
Total (*n*)	40	23	7	5	5	40
Male: Female	7:33	19:4	1:6	3:2	3:2	27:13
Age (years)	60 (42–84)	54 (28–76)	61 (32–76)	56 (47–73)	61 (45–78)	62 (39–82)
Subtype						
iCCA	-	-	-	-	-	27
pCCA	-	-	-	-	-	12
dCCA	-	-	-	-	-	1
CA19-9 (U/mL)	8.2	12.6	4	9.2	7.4	351.2
	(2–29)	(0.6–1000)	(0.6–168)	(0.6–555)	(0.6–1000)	(0.6–1000)
CA19-9 > 37 U/mL (%)	0/40	1/23	2/7	2/5	2/5	28/40
	(0%)	(4%)	(29%)	(40%)	(40%)	(70%)

Median values are given with (range). Abbreviations: iCCA = intrahepatic cholangiocarcinoma (CCA), pCCA = perihilar CCA, dCCA = distal CCA.

**Table 2 diagnostics-11-00589-t002:** The correlation of clinicopathological data and serum levels of candidate biomarkers in CCA patients.

Variables	S100A9 (ng/mL)			MUC5AC (ng/mL)			Angiopoietin-2 (pg/mL)			TGF-β1 (ng/mL)			CA19-9 (U/mL)		
	OR ^a^	Mean ± S.D. (*n*)	*p* Value	OR^a^	Mean ± S.D. (*n*)	*p* Value	OR ^a^	Mean ± S.D. (*n*)	*p* Value	OR ^a^	Mean ± S.D. (*n*)	*p* Value	OR ^a^	Mean ± S.D. (*n*)	*p* Value
**Age (Year)**															
<61	-	381.95 ± 231 (*n* = 18)	0.807	-	144.62 ± 108 (*n* = 18)	0.17	-	2211.14 ± 1320 (*n* = 18)	0.634	-	41.62 ± 16 (*n* = 27)	0.836	-	501.67 ± 458 (*n* = 18)	0.946
≥61	1.042	397.19 ± 257 (*n* = 22)		0.441	116.12 ± 121 (*n* = 22)		1.667	2323.84 ± 2086 (*n* = 22)		0.974	40.74 ± 16 (*n* = 29)		0.571	455.28 ± 433 (*n* = 22)	
**Gender**															
Female	-	315.75 ± 178 (*n* = 13)	0.22	-	97.16 ± 56 (*n* = 13)	0.479	-	2042.81 ± 679 (*n* = 13)	0.87	-	41.64 ± 15 (*n* = 18)	0.88	-	357.23 ± 393 (*n* = 13)	0.549
Male	1.486	426.24 ± 264 (*n* = 27)		1.256	144.25 ± 133 (*n* = 27)		0.686	2384.02 ± 2099 (*n* = 27)		0.618	40.94 ± 32 (*n* = 38)		3.59	533.41 ± 456 (*n* = 27)	
**Histological types**															
Non-Papillary	-	396.62 ± 261 (*n* = 26)	1	-	128.11 ± 119 (*n* = 26)	0.681	-	2457.98 ± 2084 (*n* = 26)	0.66	-	39.08 ± 15 (*n* = 33)	0.24	-	472.40 ± 435 (*n* = 26)	0.967
Papillary	2.133	378.66 ± 213 (*n* = 14)		1	130.50 ± 111 (*n* = 14)		0.758	1929.81 ± 878 (*n* = 14)		1.146	44.16 ± 16 (*n* = 23)		1.023	483.13 ± 463 (*n* = 14)	
**Recurrence**															
No	-	382.94 ± 248 (*n* = 23)	0.795	-	140.03 ± 133 (*n* = 23)	0.662	-	2294.07 ± 1187 (*n* = 23)	0.194	-	39.54 ± 13 (*n* = 34)	0.342	-	494.04 ± 446 (*n* = 23)	0.935
Yes	1.75	400.33 ± 243 (*n* = 17)		0.952	113.94 ± 86 (*n* = 17)		0.709	2244.78 ± 2374 (*n* = 17)		1.382	43.68 ± 19 (*n* = 22)		0.91	451.99 ± 443 (*n* = 17)	
**TNM stages**															
I, II	-	311.73 ± 173 (*n* = 9)	0.377	-	104.59 ± 33 (*n* = 9)	0.771	-	1453.78 ± 820 (*n* = 9)	0.020 *	-	32.65 ± 9 (*n* = 11)	0.045 *	-	507.11 ± 434 (*n* = 9)	0.924
III, IV	1.029	413.15 ± 258 (*n* = 31)		1.667	136.02 ± 129 (*n* = 31)		4.846 **	2510.10 ± 1897 (*n* = 31)		5.333 **	43.25 ± 16 (*n* = 45)		0.505	467.17 ± 447 (*n* = 31)	
**Metastasis**															
No	-	361.12 ± 251 (*n* = 17)	0.436	-	96.25 ± 34 (*n* = 17)	0.404	-	2262.01 ± 2389 (*n* = 17)	0.268	-	35.30 ± 15 (*n* = 22)	0.024 *	-	462.92 ± 451 (*n* = 17)	0.892
Yes	1.31	411.93 ± 240 (*n* = 23)		1.857	153.11 ± 145 (*n* = 23)		1.857	2281.34 ± 1164 (*n* = 23)		3.467 **	44.96 ± 15 (*n* = 34)		0.723	485.93 ± 440 (*n* = 23)	
**Lymph node metastasis**															
No	-	388.94 ± 271 (*n* = 18)	0.765	-	99.57 ± 36 (*n* = 18)	0.644	-	2218.06 ± 2325 (*n* = 18)	0.201	-	36.79 ± 15 (*n* = 25)	0.062	-	437.24 ± 451 (*n* = 18)	0.545
Yes	1.042	391.47 ± 224 (*n* = 22)		1.5	152.98 ± 149 (*n* = 22)		2.27	2318.18 ± 1178 (*n* = 22)		3.111 **	44.69 ± 16 (*n* = 31)		0.865	507.99 ± 438 (*n* = 22)	
**Distant metastasis**															
No	-	387.51 ± 239 (*n* = 34)	0.88	-	124.49 ± 117 (*n* = 34)	0.225	-	2236.04 ± 1816 (*n* = 34)	0.544	-	40.85 ± 17 (*n* = 47)	0.739	-	509.19 ± 444 (*n* = 34)	0.197
Yes	1.267	406.31 ± 290 (*n* = 6)		1.2	154.20 ± 108 (*n* = 6)		2.25	2483.27 ± 1550 (*n* = 6)		2.827	42.79 ± 9 (*n* = 9)		0.225	288.93 ± 392 (*n* = 6)	

The symbol (*) indicates a statistically significant *p* value < 0.05, as determined by 2 independent sample t-tests. ^a^ Odd ratios (OR) were analyzed to demonstrate the association of serum levels of candidate biomarkers with clinicopathological variables. The symbol (-) indicates a reference variable and symbol (**) denotes a statistically significant *p* value < 0.05, as analyzed by logistic regression.

**Table 3 diagnostics-11-00589-t003:** The performance of single biomarker for CCA diagnosis, based on the best cut-off derived from ROC analysis and YI.

Group Comparisons	Biomarkers	Cut-Off	AUC (95% CI)	YI	SN	SP	LR	*p* Value
Normal vs. CCA	S100A9 (ng/mL)	>197.9	0.888 (0.818–0.958)	0.7	77.5	87.5	6.2	<0.0001
	MUC5AC (ng/mL)	>104.6	0.639 (0.517–0.762)	0.3	52.5	77.5	2.3	0.032
	TGF-β1 (ng/mL)	>33.42	0.649 (0.535–0.762)	0.3	76.8	57.5	1.8	0.013
	Angiopoietin-2 (pg/mL)	>2422	0.567 (0.439–0.695)	0.2	35	87.5	2.8	0.303
	CA19-9 (U/mL)	>23.34	0.768 (0.644–0.893)	0.7	72.5	97.5	29	<0.0001
Normal vs. Non-CCA	S100A9 (ng/mL)	>197.9	0.832 (0.735–0.928)	0.6	72.5	87.5	5.8	<0.0001
	MUC5AC (ng/mL)	>128	0.545 (0.417–0.672)	0.2	32.5	82.5	1.9	0.492
	TGF-β1 (ng/mL)	>22.81	0.618 (0.495–0.741)	0.2	95	27.5	1.3	0.068
	Angiopoietin-2 (pg/mL)	<1312	0.530 (0.398–0.662)	0.3	42.5	82.5	2.4	0.644
	CA19-9 (U/mL)	>23.50	0.573 (0.442–0.703)	0.3	32.5	97.5	13	0.264
Non-CCA vs. CCA	S100A9 (ng/mL)	>87.11	0.525 (0.398–0.653)	0.1	97.5	15	1.1	0.697
	MUC5AC (ng/mL)	>90.51	0.581 (0.454–0.708)	0.3	65	60	1.6	0.213
	TGF-β1 (ng/mL)	>39.98	0.551 (0.432–0.671)	0.2	62.5	60	1.6	0.391
	Angiopoietin-2 (pg/mL)	>1008	0.581 (0.455–0.708)	0.2	90	32.5	1.3	0.211
	CA19-9 (U/mL)	>37.39	0.716 (0.595–0.837)	0.5	70	82.5	4	0.0009

Abbreviations; AUC = area under the ROC curve, YI = Youden index, SN = sensitivity, SP = specificity, LR = likelihood ratio. The *p* value < 0.05 was considered statistically significant.

**Table 4 diagnostics-11-00589-t004:** The predictive risk of CCA and other cancers relative to normal control group by using serum levels of candidate biomarkers panel.

Comparative Diagnosis	Biomarkers	Crude	*p* Value	Adjusted	*p* Value
		OR (95% CI)		OR * (95% CI)	
Normal vs. CCA	S100A9 < 197.9 vs. ≥ 197.9 ng/mL	24.11 (7.30–79.68)	<0.0001	22.50 (5.77–87.82)	<0.0001
	MUC5AC < 104.6 vs. ≥ 104.6 ng/mL	3.81 (1.45–10.02)	0.007	3.78 (1.23–11.57)	0.02
	TGF-β1 < 33.42 vs. ≥ 33.42 ng/mL	3.16 (1.25–7.93)	0.015	2.84 (0.97–8.28)	0.055
	Angiopoietin-2 < 2422 vs. ≥ 2422 pg/mL	3.77 (1.21–11.79)	0.023	5.17 (1.35–19.78)	0.016
	CA19-9 < 23.34 vs. ≥ 23.34 U/mL	102.82 (12.56–841.96)	<0.0001	129.44 (12.90–1295.60)	<0.0001
Normal vs. Non-CCA	S100A9 < 197.9 vs. ≥ 197.9 ng/mL	18.46 (5.75–59.23)	<0.0001	30.95 (6.79–141.05)	<0.0001
	MUC5AC < 128 vs. ≥ 128 ng/mL	2.27 (0.79–6.49)	0.126	2.08 (0.63–6.88)	0.23
	TGF-β1 < 22.81 vs. ≥ 22.81 ng/mL	7.21 (1.48–35.06)	0.014	4.98 (0.91–27.31)	0.064
	Angiopoietin-2 > 1312 vs. ≤ 1312 pg/mL	3.48 (1.25–9.75)	0.017	1.30 (0.36–4.72)	0.694
	CA19-9 < 23.50 vs. ≥ 23.50 U/mL	18.78 (2.32–152.16)	0.006	40.13 (4.17–385.64)	0.001
Non-CCA vs. CCA	S100A9 < 87.11 vs. ≥ 87.11 ng/mL	6.88 (0.79–60.06)	0.081	8.04 (0.87–74.47)	0.066
	MUC5AC < 90.51 vs. ≥ 90.51 ng/mL	2.79 (1.13–6.90)	0.027	3.26 (1.25–8.52)	0.016
	TGF-β1 < 39.98 vs. ≥ 39.98 ng/mL	1.83 (0.75–4.45)	1.81	0.143 (0.79–5.07)	2.004
	Angiopoietin-2 < 1008 vs. ≥ 1008 pg/mL	4.33 (1.27–14.78)	0.019	4.78 (1.35–16.96)	0.015
	CA19-9 < 37.39 vs. ≥ 37.39 U/mL	11 (3.81–31.73)	<0.0001	12.7 (4.00–40.31)	<0.0001

* Odds ratio adjusted for age and sex statistical analysis. Abbreviations; OR: odds ratio, CI: confidence interval, CCA: Cholangiocarcinoma. The *p* value < 0.05 was considered statistically significant.

**Table 5 diagnostics-11-00589-t005:** The best tree’s parameters for each decision tree (DT) model after performing five-fold cross validation method with *GridSearchCV* criterion.

Parameter	DT I (Normal vs. CCA)	DT II (Normal vs. Non-CCA)	DT III (Non-CCA vs. CCA)
max_depth	9	9	3
max_feature	4	1	2
min_samples_leaf	3	3	3
min_samples_split	10	14	10
criterion	‘gini’	‘gini’	‘gini’

**Table 6 diagnostics-11-00589-t006:** Classification performance of five single biomarkers and DT models for three comparative diagnosis.

Comparative Diagnosis	Single Biomarkers and DTs	Classification Performance					
		SN	SP	YI	PPV	NPV	ACC
Normal vs. CCA	S100A9	86	80	0.7	86	80	83
	MUC5AC	43	90	0.3	86	53	63
	TGF-β1	71	60	0.3	71	60	67
	Angiopoietin-2	14	**100**	0.1	**100**	45	50
	CA19-9	71	**100**	0.7	**100**	71	83
	DT I: CA19-9 and S100A9	**93**	80	**0.7**	87	**89**	**88**
Normal vs. Non-CCA	S100A9	67	83	0.5	**80**	71	75
	MUC5AC	33	67	0	50	50	50
	TGF-β1	**100**	17	0.2	55	**100**	58
	Angiopoietin-2	50	83	0.3	75	63	67
	CA19-9	33	**92**	0.3	**80**	58	63
	DT II: angiopoietin-2, TGF-β1, and S100A9	92	67	0.6	73	89	**79**
Non-CCA vs. CCA	S100A9	**100**	23	0.2	52	**100**	58
	MUC5AC	91	54	0.5	63	**88**	71
	TGF-β1	36	54	0	40	50	46
	Angiopoietin-2	82	62	0.4	64	80	71
	CA19-9	73	**100**	0.7	**100**	81	**88**
	DT III: TGF-β1, CA19-9, angiopoietin-2, and MUC5AC	82	92	**0.7**	**90**	86	**88**

The bold indicates the highest value, while the underline indicates the second-highest value in each category of biomarker diagnostic performance. Abbreviations; SN = sensitivity; SP = specificity; YI = Youden index; PPV = positive predictive value; NPV = negative predictive value; ACC = accuracy.

## Data Availability

The data underlying this article are available in the article, in its online supplementary materials, and from the corresponding author upon reasonable request.

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
