# Peer review of "Establishment of a Potential Serum Biomarker Panel for the Diagnosis and Prognosis of Cholangiocarcinoma Using Decision Tree Algorithms"

_diagnostics, 2021, doi:10.3390/diagnostics11040589_

Round 1
Reviewer 1 Report
We reviewed with a great interest the manuscript by Kimawaha and colleagues about the identification of a novel combination of serum biomarkers aiming at improving the diagnosis and prognosis of cholangiocarcinoma (CCA). In this study, the authors identified a panel of potential serum biomarkers specific for CCA patients using a decision tree construction. Thus, levels of S100A9, MUC5AC and TGF-b1 were enhanced in CCA serums when compared to healthy individuals. In addition, these serum biomarkers were also able to discriminate CCA from other gastrointestinal tumors tract tumors. In addition, the expression of TGF-b1 and angiopoietin-2 were both correlated with a poor prognosis for CCA patients.
Altogether these data bring some interesting points supporting the use of liquid biopsies combined with bioinformatics approaches as an innovative tool to improve the diagnosis and prognosis of CCA. However, these data bring little added value to CA19-9 based diagnosis approaches. Several comments and suggestions below may help improving this study.
Major points
1- The validation of the biomarkers panel in an external bank of liquid biopsies is mandatory to support the conclusion of this study.
2- Serum extracellular vesicles were shown to be enriched in protein biomarkers. By measuring the serum levels of their panel of proteins into these vesicles, the authors could improve their specificity and sensitivity scores.
3- The authors should distinguish extrahepatic CCA (eCCA) and intrahepatic CCA (iCCA). eCCA and iCCA are two distinct entities characterized by specific molecular alterations and specific lines of treatment. Thus, diagnosis tools should be specific for either eCCA or iCCA.
4- The authors should investigate whether the serum levels of the biomarker panel are associated with the relapse free survival in CCA patients.
5- A section showing the potential of the biomarkers panel as prognosis tool should be added in the abstract to be in line with the title.
Minor points
line 47: “identify” should be ”differentiate”
line 65: “The meta-analysis” should be “A meta-analysis”
Figure 3: enlarge the figure (barely readable)
Author Response
Dear Reviewer
We deeply appreciate all reviewers for critical reading of our manuscript entitled “Establishment of a potential serum biomarker panel for the diagnosis and prognosis of cholangiocarcinoma using decision tree algorithms”, by Kimawaha et al. We thank the reviewer for comment and suggestion. We have revised and responded to each comment point by point in the “Response to Reviewer”, please see the attached file.
We believe that the manuscript has been improved satisfactorily and hope that it is acceptable for publication in the Diagnostics (ISSN 2075-4418).
Yours Sincerely,
Anchalee Techasen, Ph.D.
On behalf of all authors

Reviewer 2 Report
This paper constructed decision tree (DT) models to classify cholangiocarcinoma (CCA) from the healthy subjects and gastrointestinal cancer patients based on the 4 CCA biomarkers, S100 calcium binding protein A9 (S100A9), mucin 23 5AC (MUC5AC), transforming growth factor β1 (TGF-β1), angiopoietin-2 and carbohydrate antigen 19-9. This is an interesting paper.
The decision trees were constructed based on the Classification and Regression Tree (CART) algorithm. Although you have cited a reference about Classification and Regression Tree (CART) algorithm. It is not very clear about the Python code and the functions of the 5 parameters, max_depth, max_feature, min_samples_leaf, 153 min_samples_split, and criterion. More descriptions of CART are needed.
Author Response

(The authors gave the same response as above.)

Reviewer 3 Report
The manuscript by Kimawaha et al. is mostly focused on the establishment of a potential serum biomarker panel for the diagnosis and prognosis of cholangiocarcinoma. The authors analyzed the serum levels of 5 different potential biomarkers and evaluated the potential diagnostic of them, both isolated and combined, further comparing with the levels of CA19-9. The authors evaluated the levels in patients with CCA and compared with healthy individuals and patients without CCA (HCC + gallbladder cancer + pancreatic cancer). The authors herein nicely reported that the serum levels of S100A9, MUC5AC and TGF-beta were markedly increased in patients with CCA, presenting great diagnostic capacity values both for diagnosis of CCA when compared with healthy or with individuals with non-CCA tumors. However, these biomarker candidates did not present a superiority over CA-19.9 capacity when used alone. Outstandingly, the combination of some of them provided excellent diagnostic values. The work was well-conducted and the manuscript is well-written and contain very important and clinically sound results in the field. There are only minor points that should be addressed in order to improve the already great quality of this work.
1- Et al should be presented in italics
2- What is the rationale behind the selection of these 5 specific biomarkers?
3- In table 1, please provide the data regarding the subtype of CCA (according to the localization presence of PSC), and the etiology of the tumors (% of cirrhosis, viral, NAFLD, alcohol, etc) for all the tumors.
4- In table 2, please include the Odds Ratio values.
5- In figure 3, it is possible to read the legend of the colors. Please increase the size.
6 – It would be very interesting to evalute if the predicte panel specifically help in the differential diagnosis of HCC vs iCCA and dCCA vs pancreatic cancer.
Author Response

(The authors gave the same response as above.)

Round 2
Reviewer 3 Report
All my questions and concerns were properly addressed and the manuscript was improved. It is now ready to be published. Congratulations on this great work.